# Laser-Engraved Liquid Metal Circuit for Wearable Electronics

**DOI:** 10.3390/bioengineering9020059

**Published:** 2022-01-30

**Authors:** Shuting Liang, Xingyan Chen, Fengjiao Li, Na Song

**Affiliations:** 1College of Chemical and Environmental Engineering, Chongqing University of Arts and Sciences, Chongqing 402160, China; cxy525969790@gmail.com; 2Chongqing Key Laboratory of Environmental Materials & Remediation Technologies, Chongqing University of Arts and Sciences, Chongqing 402160, China; 3Shenzhen Automotive Research Institute, Beijing Institute of Technology, Shenzhen 518118, China; lifengjiao@szari.ac.cn; 4Department of Oncology, Chongqing Municipal Chinese Medicine Hospital, Chongqing 400021, China; songna896600@163.com

**Keywords:** liquid metal, laser engraving, flexible electronics, wearable electronics

## Abstract

Conventional patterning methods for producing liquid metal (LM) electronic circuits, such as the template method, use chemical etching, which requires long cycle times, high costs, and multiple-step operations. In this study, a novel and reliable laser engraving micro-fabrication technology was introduced, which was used to fabricate personalized patterns of LM electronic circuits. First, by digitizing the pattern, a laser printing technology was used to burn a polyethylene (PE) film, where a polydimethylsiloxane (PDMS) or paper substrate was used to produce grooves. Then, the grooves were filled with LM and the PE film was removed; finally, the metal was packaged with PDMS film. The experimental results showed that the prepared LM could fabricate precise patterned electronic circuits, such as golden serpentine curves and Peano curves. The minimum width and height of the LM circuit were 253 μm and 200 μm, respectively, whereas the printed LM circuit on paper reached a minimum height of 26 μm. This LM flexible circuit could also be adapted to various sensor devices and was successfully applied to heart rate detection. Laser engraving micro-processing technologies could be used to customize various high-resolution LM circuit patterns in a short time, and have broad prospects in the manufacture of flexible electronic equipment.

## 1. Introduction

Research on flexible wearable technologies and wearable electronic systems has been developing for a long time. Flexible electronic devices are characterized by their softness, stretchability, and wear resistance, and have been widely used in biomedical testing, electronic skin sensors, smart skins, photoelectricity, energy storage, and other fields [1,2,3,4,5,6]. Wearable electronic skin generally has sensors, such as stress, temperature, light, and electrochemical sensors, to measure the skin epidermis or superficial tissues.

Electronic skin is made of conductive and stretchable materials, which provide compliance for the manufacture of flexible wearable devices. Flexible circuits have been generally based on metal nanoparticles, nanowires, and graphene [7]. Compared with common organic or inorganic conductive materials, such as silver nanoparticles, liquid metal (LM) has a similar conductivity and adhesion to nano silver, and possesses the functions of stretching, bending, and torsion [8].

LM (i.e., EGaIn) is a liquid at room temperature (20 °C) and normal pressure (101.325 kPa), and has great potential in flexible electronics, catalysts, and robots [9,10,11]. Unlike mercury, which is toxic, gallium-based liquid metals are non-toxic and bio-compatible [12,13]. As a conductive material, LM is characterized by its high stability, high surface tension [14], high density, high conductivity, flexibility, viscosity, and non-toxicity [15,16]. Due to its good mobility and conductivity, GaIn alloy is regarded as a flexible conductor [17]. LM alloys based on gallium are promising flexible wearable materials with excellent mobility, strong compliance, environmental friendliness, and easy recovery. LM technology has been comprehensively developed and applied for printing electronics [18,19], sensors [20], flexible machinery [21], and 3D printing [22,23].

Flexible wearable devices made by combining LM with sensors can be used for a long time without bringing discomfort to users and can be manufactured on a large scale in practical applications. LM has been combined with different sensing fibers or sensors for real-time health monitoring; these combinations could be used as promising wearable belt platforms.

With the progress of studies and applications of LM skin in recent years, a general LM circuit manufacturing method has been developed to solve the problems of circuit manufacturing and printing. Various LM pattern techniques have been developed in the past few years based on desktop 3D printing [9,24], liquid phase 3D printing [25], compatible hybrid 3D printing [26], suspension 3D printing [27], stencil lithography [28,29], inkjet printing [30], fused deposition printing [31], micro-contact printing [32], dual-trans printing [33], micro-fluidic injection [34,35], and selective liquid-metal plating (SLMP) [36].

However, these reported techniques have involved multiple-step operations (such as 3D printing), additional pre-treatment of the substrate (such as stencil lithography), post sintering (such as fused deposition printing), delicate molds and masks (such as inkjet printing), tedious microfabrication processes (such as microfluidic injection), alongside sophisticated equipment. These studies have not only complicated the fabrication process but also increased the cost.

In the past, the circuit pattern manufactured by the template method had a certain precision, but the variability was low and the manufacturing process was complicated. Polydimethylsiloxane (PDMS) grooves filled with LM have previously been reported [37,38]. Hand-drawn patterning could solve the problem of individualization, but could not meet the requirements for preparing circuits. The resulting pattern was not uniform and could not be used for industrial production.

These problems have restricted the further development and application of LM-based materials; thus, it is particularly urgent and necessary to solve the problem of fine printing. Traditional LM circuit preparation has been challenging to develop further. Laser technology has been widely used in various fields, for example: textile electronic [39], composite materials [40], biomedical science [41,42,43], and so on. To the best of our knowledge, there have been few reports on the fabrication of ultra-micro LM electronic devices by laser engraving [44,45]. For example, Pan et al. used laser printing to prepare a transparent conductive film, but they used a laser to print directly onto the plane of the LM, which caused the burning and deterioration of the LM [46].

In this study, we proposed and demonstrated a new printing and manufacturing strategy for LM electronic circuits, denoted as laser printing-based LM electronic circuit printing. Laser engraving on a PDMS substrate and paper templates were used to produce LM flexible electronic circuits and electronic products. For illustration, flexible LM electronic products were applied on PDMS and paper-based materials. The accuracy of the circuit could be easily adjusted by the laser parameters, and the method was also compatible with a variety of substrates. Finally, the application of the patterned LM circuit as a heart rate sensor was demonstrated.

## 2. Experimental Section

### 2.1. Materials Fabrication

First, the indium metal (25.5 g) was cut into small pieces and weighed. A straw was used to extract liquid gallium (84.5 g), which was then mixed with the indium metals. After that, the mixture was gently put into a magnetic stirrer with a heating function, and the temperature was set at 68 °C for 30 min. Care was taken not to heat the liquid alloy for 5 h to avoid failure.

The preparation of the PDMS substrate involved the following steps: First, in the mold, the base and curing agent were evenly applied in a uniform rotary coating at the mass ratio of 10:1. Next, the coating was held at room temperature under irradiation with ultraviolet light for approximately 12 h, to obtain a stretchable transparent PDMS substrate. To ensure the cleanliness of the PDMS substrate and facilitate the printing of LM circuits, an ultra-thin viscous polyethylene (PE) film was placed on the surface of the PDMS substrate.

### 2.2. Laser Engraving Process

As a processing medium, laser engraving is based on laser numerical control technology. The instant melting and gasification physical denaturation of the processing materials under laser irradiation achieves the purpose of processing.

The manufacturing process of the laser engraved LM circuit was as follows. The laser engraving machine (DOBOT MOOZ, product model: DT-MZ-2ZFU-00E, working range X130*Y130 mm, laser power 0.5 W) was purchased from Chongqing Expansion Electronics Co., Ltd. (Chongqing, China). The laser type was solid YAG, and the wavelength was 355 nm.

Some clear, well-defined pattern pictures were imported into the computer software MoozStudio, which were then configured by the laser engraving machine. Then, the software generated a GCODE file that was recognizable to the laser engraving machine. Then, the file was output to a portable mobile device with a USB interface, which was connected to the laser engraver. The *x*, *y*, and *z*-axis coordinates were adjusted so that the laser could engrave to the horizontal plane of the substrate. On the vertical axis, objects could be removed by laser printing. However, it was difficult to measure the laser machining rate along the vertical axis with the available equipment.

The laser engraving machine engraved the set circuit according to the set pattern. The maximum engraving rate was 5 mm/s, the maximum laser movement rate was 8.333 mm/s, and the laser power was 0~100%. Typically, after the laser printing started, 3–4 s was required to reach the set engraving speed. For instance, if the carving speed was 5 mm/s, then the acceleration rate of the stages for engraving was 1.667 mm/s^2^. We generally did not engrave PE-PDMS or paper during the acceleration stage. When the speed of laser printing became a stable, we began to print PE-PDMS and paper.

The printing process of the LM circuit was as follows. After printing the circuit onto the PE-PDMS substrate, a thin tip with a diameter of only 1 mm was used to take LM (5 mg), and evenly apply it along the circuit to fill the engraved line. Then, the PE film attached to the PDMS was gently peeled off to obtain a continuous, complete LM circuit. Finally, the prepared PDMS was coated onto the LM and PDMS substrate.

For paper-based printing, the laser-engraved pattern template was placed on a piece of paper as a base. Using a fine nib (1 mm in diameter), 5 mg of LM was extracted. For the paper-based LM circuits, instead of using PE as a template, we used a piece of paper as a template. The LM was passed through the groove of the paper template line and was printed onto the next layer of paper, resulting in a fine LM line. After filling the paper substrate with LM, the paper-based LM circuit was bared.

### 2.3. Electrical/Morphological Characterization

Characterization of the laser-engraved LM circuit was performed as follows. The size and shape of the LM circuits that were printed on the paper and PDMS substrates were obtained using a scanning electron microscope (SEM; ZEISS, SIGMA 300) with a working distance (WD) of 4.1–5.8 mm, an EHT acceleration voltage of 1–3 kV, a magnification of 28–440×, and a ln-Lens detector. The cross-sectional morphology of the laser-engraved groove could be also obtained using SEM.

To measure the electrical properties during mechanical deformation, a high-precision digital multimeter (Victory VC9808, Sheng-Sheng Sheng-Li Technology Co., Ltd., Shenzhen, China) was used to measure the resistance characteristics of the circuit under bending, stretching, and torsion.

For demonstrating an application based on a laser-engraved LM circuit sensor, a pulse heart rate measurement photoelectric reflection type analog sensor (i.e., a Pulse sensor) was connected to the Arduino development board (Uno R3) through the laser printing LM circuit. The pulse sensor was marked with an S signal output line and the Arduino Analog input A0 was connected as follows: + was connected to 5 V, and − was connected to GND. The Arduino was connected to the USB interface on the computer through the square port data cable. The process of displaying the pulse image was programmed by the Arduino.

## 3. Results and Discussion

The manufacturing steps of the laser-engraved LM circuit are shown in Figure 1a. First, a thin PE film was used to wrap the PDMS substrate, (Figure 1a). Then, the laser engraving machine started engraving according to the set circuit pattern. The pattern of the laser engraving circuit was set using the computer software MoozStudio. The obtained output GCODE file was copied to a portable mobile device and connected to the laser engraving machine.

Both the PE film and PDMS were able to absorb the laser energy. The laser could break through the PE film with a thickness of 0.5 mm. However, for PDMS with a thickness of 3–5 mm, the laser only left a groove in the surface. After the engraving was completed, the burning PE and PDMS materials in the groove were gently wiped off with a metal needle, and a circuit pattern with a set circuit path was obtained.

Then, a fine brush was used to pick up the configured LM, which was then slowly applied to the laser-engraved groove. This fine brush had a tip with a diameter of only 1 mm. Moreover, spraying was also used to fill the groove with LM.

The function of the PE film was to adhere to the PDMS substrate such that it would not fall off during printing; the PE film could also effectively prevent external dust from polluting the PDMS substrate. More importantly, the laser penetrated the PE film and part of the PDMS substrate in the process of printing the circuit. Therefore, in the process of filling the groove with LM, the LM entered the groove in the PDMS substrate through the ablated PE film to prepare a complete LM circuit. The non-punctured PE film (i.e., the part that did not require printing) effectively prevented the PDMS substrate from contacting the LM, by preventing the LM from moving outside of the groove carved by the laser.

Finally, the PE film attached to the PDMS was gently peeled off to obtain a continuous, complete LM metal circuit. After completion of the LM filling, the PE film was easily torn off to obtain a complete LM circuit on PDMS. Thereafter, the prepared PDMS film was coated on the LM and PDMS substrate.

This packaging process did not affect the integrity and resolution of the laser-engraved pattern. After sealing the device, the resulting laser-engraved LM flexible circuit could be used in a variety of applications, such as flexible electronic skin.

Laser processing has some excellent features, such as: no contact with the material surface, no dependency on mechanical movement, and generally no need to be fixed. Simultaneously, laser engraving also has the beneficial characteristics of high machining accuracy, high speed, and wide application prospects [47]. Laser engraving is not affected by the material elasticity and flexibility; therefore, it is also convenient for processing soft materials. The direct laser engraving system mainly consisted of three parts: a high-energy laser, a laser delivery system, and an optical system (Figure 1b–d).

To better demonstrate the surface patterning capabilities of the laser-engraved circuits on PDMS, we subsequently used laser engraving to create more complex patterns, which also verified the feasibility of laser engraving for the fabrication of flexible electrons. A variety of complex electronic patterns were successfully produced, as shown in Figure 2a–d. These patterns had sharp features with high precision and resolution.

Gold serpentine and Peano curve patterns were obtained by laser engraving, as shown in Figure 2c,d. A magnification pattern with a high quality of patterning of the Peano-based wire is shown in Figure 2e, which emphasized the feasibility of the laser engraving method. No defects of corners and turns were observed.

Reducing the thickness of the PDMS-based electronic devices is also important for achieving highly flexible electronic devices. These highly flexible, mechanically durable patterns were easy to manufacture, which was a step towards flexible circuits. Thus, we also studied the microscopic characteristics of the laser-engraved LM wires and patterns on PDMS and paper substrates. Figure 3a–d shows the micrographs of the laser engraved LM circuits on PDMS and paper bases.

The cross-sectional SEM image of the PDMS-based LM is shown in Figure 3a. The width of the groove printed by laser was approximately 440 μm, and the depth was approximately 200 μm. The LM was filled into the laser-engraved trench. The laser-engraved trench had no obvious defects on the micron level. Since the groove was formed by burning PDMS with laser, it had a certain roughness. However, the width of the grooves remained uniform at a uniform laser rate. The wettability of LM on PDMS has been reported in a previous article [48]. LM forms a thin oxide layer on its surface in the presence of oxygen. This oxide shell easily adheres to the surface of almost any material, including PDMS substrates.

Figure 3b shows an SEM image of an LM line that was filled on a laser-engraved PDMS substrate. The LM covered and filled the trench. The edges of the LM and laser-engraved trenches were snugly fit and maintained a stable surface topography.

However, a small hole existed in the edge, similar to a pore, as seen in Figure 3b, which was due to the surface tension and oxidation of LM. In fact, the existence of the small hole was due to the oxidized LM not adhering to the edge of the groove. When the mixture of Ga-In alloys was oxidized, the fluidity of the LM was poor. To prevent such small holes, the experiments should be performed as soon as possible to prevent the oxidation of large areas of LM.

The width of the LM was approximately the same as the width of the laser-engraved groove, which indicated that the LM had good adhesion in the groove. The LM was observed to be tightly connected. There was no visible fracture, which ensured the connection of the circuit. This proved the feasibility and accuracy of laser engraving for manufacturing flexible circuits.

As seen from Figure 3c,d, the LM circuit printed on the paper base had a cross-sectional height of approximately 26 μm and a width of approximately 253 μm. The surface of this LM circuit was smooth and flat, and uniformly covered the paper base without breakage. The laser-engraved printed LM circuit could be applied to an ultra-thin flexible circuit. Figure 3e shows an SEM image of an LM line at different magnifications printed on a paper base. The LM circuits printed by laser engraving appeared as a smooth continuous straight line on the paper base. Through the high-magnification observation of the LM and paper interface, the interface between the printed LM circuit and paper was clear with a favorable resolution.

To achieve a certain carving effect on a specific material, it was required to absorb a certain amount of laser energy, regarded as the laser energy absorbed. By adjusting the focal length, the laser energy per unit area could also be adjusted. A high-speed laser head resulted in high productivity. From the above conditions, the laser energy absorbed by the material = laser power (W)/engraving speed (mm/s). The energy absorbed by the material is given by:(1)$$E_{materia}=\frac{J}{s}$$
where *J* is the laser energy, *s* is the engraving time, and *E_material_* is the laser energy absorbed by the material (per unit length). The engraving speed refers to the speed at which the laser head moves, usually expressed in inches per second (IPS). 

To improve the material absorption and transmission of laser energy, the laser power should be increased, and the speed of carving should be reduced, as for the final carving effect. In the experiment, the only conditions that could be changed were the laser power and the carving speed; the laser wavelength was maintained constant at 355 nm. Therefore, dot matrix engraving could be used to scan graphics, text, and vectorized text. Dot-matrix engraving resembles high-definition dot-matrix printing. The laser head swings from side to side, carving out a line composed of a series of points in time. Then, the laser head moves up and down simultaneously to carve out multiple lines, finally forming a whole page of an image or text.

The laser head speed was also used to control the depth of the cut. For a particular laser intensity, the slower the speed, the greater the depth of cutting and engraving. To validate the effect of the laser carving speed and travel speed on the width and time of engraving on the paper base (Figure 3e–g), we changed the parameters in Moozstudio.

In Moozstudio, the default carving speed was 8.333 mm/s and the travel speed was 5 mm/s. The carving speed was changed from 1.6 to 8.333 mm/s (with a set travel speed at 5 mm/s). After obtaining this set of data, the travel speed was changed from 5 to 11.667 mm/s (with a set carving speed at 8.333 mm/s).

As shown in Figure 3f,g, when the engraving speed increased from 1.667 mm/s to 8.334 mm/s, the engraving width decreased from 650 to 253 μm. When the moving speed increased from 5 mm/s to 11.667 mm/s, the width of the engraving decreased from 300 to 200 μm. The relationship between carving speed, travel speed, carving time and engraving width in the sample was also shown in Appendix A in Appendix A.

As shown in Figure 3e, after measurement, we also found that changing the laser travel speed had little or no effect on the carving time. This was because the travel speed was the horizontal velocity of the laser emitter in the non-working state. When engraving continuous patterns, the laser transmitter was always working, so the carving time was not affected. At this time, the travel speed was set at 5 mm/s, so the total carving time did not change.

However, the carving speed had a certain influence on the carving time. The carving speed referred to the speed at which the laser spot moved, and its speed could control the cutting depth. When the material was hard, a slower carving speed could be chosen. When changing the carving speed, in this case, the laser focused on a point for a long time, causing the materials to absorb more energy. Therefore, the laser carving speed determined the time and width of the carving.

To evaluate the electrical properties of the laser-engraved LM circuits under mechanical stress conditions, the bending and tensile resistance changes were characterized. The PDMS film was filled with LM to evaluate the electrical properties of the laser-engraved circuit during mechanical deformation, where all resistance measurements were performed at least 3 times. The LM had a conductivity of 34,000 S/cm, the sheet resistance of 0.01 Ohm/sq., and stretching ability of more than 1000 [28].

First, we evaluated the resistance change of the LM wire of the Peano curve during bending. In the bending experiment, the laser-engraved LM circuit was bent at −180°, −120°, −60°, 0°, 60°, 120°, and 180°, and its resistance was measured. For each experiment, at least three sets of data were measured. As shown in Figure 4a, the resistance of the original LM wire was approximately 2.6 Ω. Similarly, there was a slight change in the relative resistance during the bending process. However, the electrical properties of the LM circuit printed by laser engraving were very stable, and the bending of the circuit was always constant at approximately zero. 

In addition to the bending test, the circuit was subjected to a torsion test with twist angles of 90°, 180°, 270°, and 360°. The laser-engraved LM circuit was fixed at both ends and, for each angle, at least three sets of data were measured. As shown in Figure 4b, the influence of the torsion angle on the electrical conductivity of the circuit was extremely small. Furthermore, the relative resistance change was approximately constant close to zero. 

After that, the laser-engraved LM circuit under a tensile state was evaluated, as shown in Figure 4c. The resistance value of the circuit increased slightly with the length of stretching in the range of 83%. A laser-engraved LM circuit with a length of 3 cm was stretched to 4 cm, 5 cm, and 5.5 cm, and the corresponding resistance was measured. After the measurement, the resistance change was less than 50%. 

For clarity in the ongoing discussion, we will now discuss the terms used to describe the electrical properties of materials. Conductivity is an inherent material property that describes how easily a current can flow through a material when a voltage is applied. Resistivity (*ρ*) is the inverse of conductivity and is a physical quantity that is used to indicate the electrical resistance and characteristics of various materials. In the case of a certain temperature, the resistance of a conductor can be expressed as: (2)$$R=\frac{\rho L}{S}$$
where *ρ* is the resistivity, *L* is the length of the material, and *S* is the area. The resistance of the material was proportional to the length of the material. That is, when the material and the cross-sectional area were constant, the longer the length, the greater the resistance of the material. The resistance of the material was inversely proportional to the cross-sectional area of the material. When the material and length were constant, the larger the cross-sectional area, and the smaller the resistance. 

The extraordinary stability of the electrical properties during mechanical deformation was due to the fluidic nature of the LM conductor [20], and the flexibility of the laser-engraved circuit. The pattern of the laser-engraved LM circuit could be easily deformed into various shapes. After relaxation, the electrical conductivity could be restored. Therefore, it was envisaged that the laser-engraved LM circuit could be flexible wearable skin electronics. The thin laser-engraved LM had very high flexibility and mechanical stability (Figure 4) in various mechanical deformations, such as stretching, torsion, and bending. 

To further demonstrate the wide application prospects of laser-engraved circuits, we designed a simple LM circuit pattern that consisted of three LM curved lines that connected the sensor to the Arduino development board. 

Using a flexible circuit, multiple functional modules could be combined, such as a sensing system, actuator, and other functional modules, which could be applied in the fields of electronic security, flexible display, and biological diagnosis. 

As shown in Figure 5a, a thin LM electronic circuit with a depth approximately 200 μm was fabricated. The circuit of the conductor and the entire working environment are shown in Figure 5b. The laser-engraved LM circuit was connected to the Arduino development board on the PDMS substrate. The pulse heart rate measurement photoelectric reflection type analog sensor (Pulse sensor) was connected to the Arduino development board (Uno R3) through a laser-printed LM circuit. The heart rate was successfully measured by the related software on the computer (Figure 5c). 

The sensor consisted of a light source and a photoelectric transducer, which were attached to the patient’s finger or earlobe by straps and clips. The photoelectric reflection method measured the pulse that was produced by different blood vessels beating in human tissue. The light source generally used a light-emitting diode, selected with a certain wavelength (from 500 nm to 700 nm), for oxygen and hemoglobin in the arterial blood. 

When the light beam passed through the peripheral blood vessels of the human body, the volume of the arterial pulsation changed, which also caused changes in the light transmittance. At this time, the light reflected by the human tissue was received by the photoelectric transducer, converted into an electrical signal, and turned into an amplified output. Since the pulse was a signal that periodically changed with the beat of the heart, the volume of the arterial vessel also changed periodically; therefore, the change period of the electrical signal of the photoelectric transducer was the pulse rate. 

The heart rate referred to the number of heartbeats in a minute (*BPM*). The clumsiest way to obtain a heart rate was to count the number of pulses in a minute. However, in this way, every heart rate measurement required a minute to obtain results, which was extremely inefficient. 

Another method was to measure the interval between two adjacent pulses (*IBI*), and then divide the interval one minute to obtain the heart rate. The heart rate sensor calculated the time interval between the two pulses using the change in the micro voltage. The advantage of this method was that the pulse could be calculated in real-time, and the efficiency was high. This led to the correlation between *IBI* and *BPM*, which can be expressed by the following formula: (3)$$BPM=\frac{60}{IBI}$$
where *IBI* is the time interval between two adjacent pulses (ms) and *BPM* is the heart rate, i.e., the number of heartbeats in a minute. Based on the above analysis, we concluded that our goal was to obtain the value of *IBI*, and then calculate the real-time heart rate (*BPM*) via the *IBI*.

Figure 5c,d were produced by the Serial Plotter tool which was provided by Arduino programming. The red curve of the pulse was the pulse wave data from the Signal variable. *IBI* was the time between each beat, whereas the *BPM* was the beats per minute. As shown in the figure, electronic skin was made by laser carving of the LM, and could measure the heartbeat in real-time. The real-time value of *IBI* was 806 ms and the *BPM* value was 78. 

To determine the usability and accuracy of the LM circuit by laser engraving, we also measured the pulse under the calm and motion states, where the curves are shown in Figure 5e,f. Figure 5e shows the image of the pulse sensor at rest, whereas Figure 5f shows the post-exercise pulse sensor image. The pulse wave data were obtained by the serial port. The experimental results demonstrated the successful integration of laser-engraved LM circuits, sensors, and electronic components, such as the Arduino development board, which enabled their practical applications. 

## 4. Conclusions

This paper introduced a simple, convenient, and reliable method for the preparation of laser engraving microfabrication technology. Using LM as a conductor, and PDMS as a substrate, a flexible circuit that had a complicated pattern with anti-deformation was manufactured. Under optimal conditions, the accuracy of the laser engraving could be used to manufacture a flexible circuit with a width of 253 μm on a PDMS substrate. These circuits were successfully applied to flexible electronic sensors, such as sensors for detecting heart rate. We envision that this laser-engraved LM flexible circuit could be combined with other state-of-the-art skin electronics, which will have a major impact on the design and construction of future devices. These findings indicated that laser engraving microfabrication technology has great potential for applications.

## Figures and Tables

**Figure 1 bioengineering-09-00059-f001:**
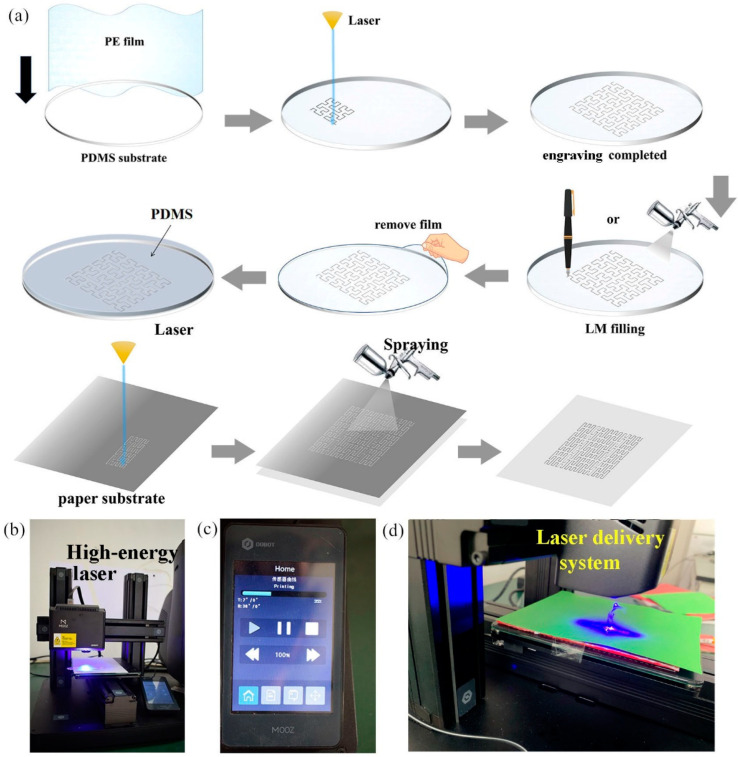
(**a**) Steps for the laser engraving of an LM circuit onto a PDMS substrate and the steps for the laser engraving of an LM circuit onto a paper substrate. (**b**) Photograph of the laser engraving of the PDMS substrate. (**c**) Display screen of the laser engraving equipment. (**d**) Photographs of the laser engraving of the paper substrate.

**Figure 2 bioengineering-09-00059-f002:**
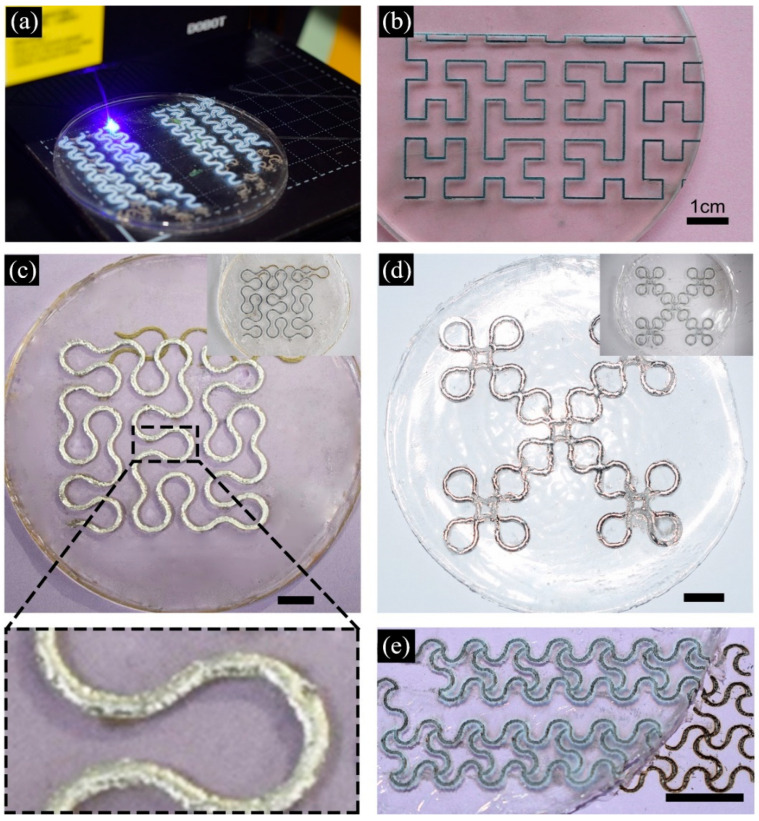
(**a**) Laser Engraving of Golden Serpentine Patterns. (**b**) Peano curve in laser engraving. (**c**) Peano-based wire before and after filling liquid metal. (**d**) Vicsek fracta on PDMS. (**e**) Gold serpentine patterns carved on paper bases and PDMS. (In (**c**–**e**), the scale is 1 cm).

**Figure 3 bioengineering-09-00059-f003:**
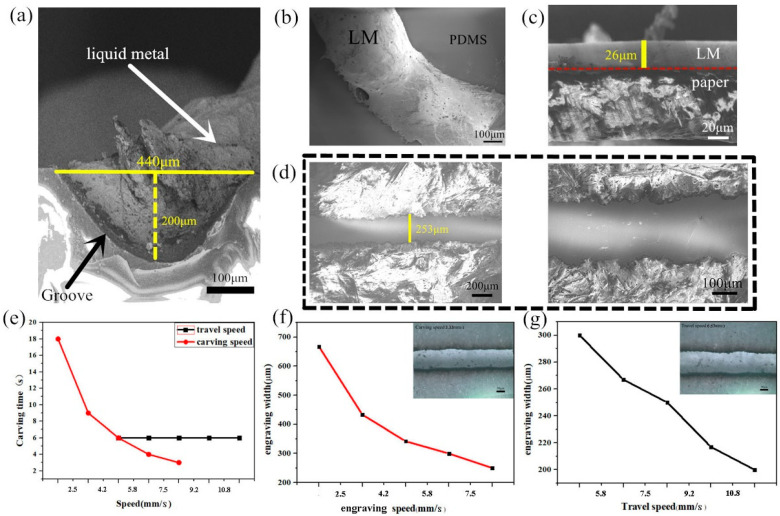
(**a**) Laser engraving of the groove pattern filled with LM on the PDMS substrate. (**b**) SEM image of the LM line filled on the PDMS substrate after laser engraving. (**c**) Cross-sectional SEM image of the LM line printed on the paper. (**d**) SEM image of the LM lines printed on the paper base at different magnifications. (**e**) Effect of carving speed and laser travel speed on the carving time. (**f**) Influence of the change in engraving speed on the width of engraving. (**g**) Influence of the change in travel speed on the width of engraving. In (**f**,**g**), the scalebar is 50 μm.

**Figure 4 bioengineering-09-00059-f004:**
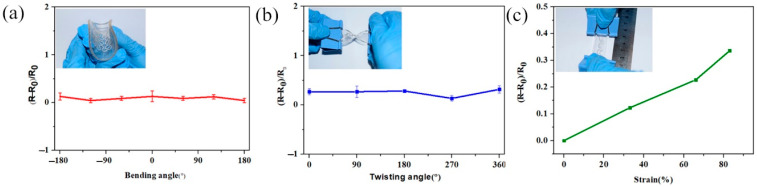
(**a**) Relative resistance changes of the LM circuit under bending. (**b**) Relative resistance changes of the LM circuit under twisting. (**c**) Relative resistance variation curves of the LM wires during tension.

**Figure 5 bioengineering-09-00059-f005:**
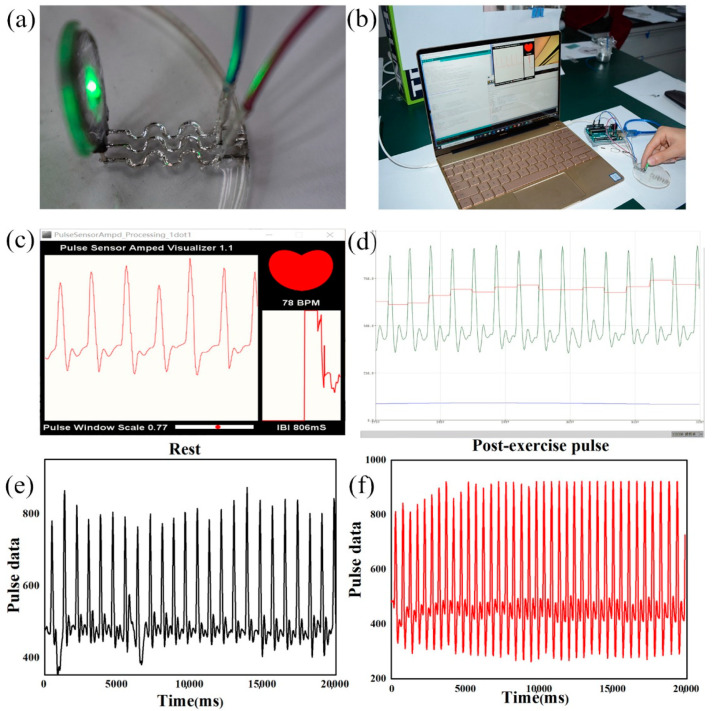
(**a**) Sensor working in LM circuits. (**b**) Flexible sensor circuit working system. (**c**) Sensor image on an Arduino sketch. (**d**) Display of sensor image on an Arduino sketch. (**e**) Pulse sensor image at rest. (**f**) Post-exercise pulse sensor image.

## Data Availability

The datasets supporting the conclusions of this article are included within the article. And the data presented in this study are available on request from the corresponding author.

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
