# Peer review of "Laser-Engraved Liquid Metal Circuit for Wearable Electronics"

_bioengineering, 2022, doi:10.3390/bioengineering9020059_

Round 1

Reviewer 1 Report

As a reviewer, I have a serious problem with this paper. The subject is timely and the laboratory work well done. In my opinion, the work could be of great interest for fellow researchers. The paper composition overall is clear. I would like to see it published.

The difficulties become apparent only when reading. Many parts of the manuscript are illegible or have no meaning despite the lack of linguistic errors. Examples are present in practically every paragraph. Long passages and short snippets. Too much for me to indicate them.

Authors should note that also text written in correct English may not have content or make no sense. And both of these possibilities were extensively used in this work.

Unfortunately, in present form, the manuscript is not suitable for publication. It must be carefully rewritten, and only then may be resubmitted. The author should bear in mind, that no language editing service can improve present text. This is a matter of the clarity of the content, not of formal linguistic correctness.

In present form, the manuscript is unreadable and not suitable for publication

Author Response

Referee 1:  As a reviewer, I have a serious problem with this paper. The subject is timely and the laboratory work well done. In my opinion, the work could be of great interest for fellow researchers. The paper composition overall is clear. I would like to see it published.  The difficulties become apparent only when reading. Many parts of the manuscript are illegible or have no meaning despite the lack of linguistic errors. Examples are present in practically every paragraph. Long passages and short snippets. Too much for me to indicate them.  Authors should note that also text written in correct English may not have content or make no sense. And both of these possibilities were extensively used in this work.  Unfortunately, in present form, the manuscript is not suitable for publication. It must be carefully rewritten, and only then may be resubmitted. The author should bear in mind, that no language editing service can improve present text. This is a matter of the clarity of the content, not of formal linguistic correctness.       

Reply: Thank you very much for your suggestions. We would like to thank the reviewer for carefully and kindly commenting the manuscript. Thank you very much for your time and feedback and valuable comments. We have closely followed your kind reminding and suggestions to further improve our paper.

We have revised the long passages and short snippets in the manuscript. The corresponding responses are highlighted in the manuscript. We also changed the clarity of the whole content. We have modified the language and content in this manuscript, which we hope meet with approval.  

Reviewer 2 Report

In the article “Laser-engraved liquid metal circuit for wearable electronics”, the authors suggested novel patterning method using laser engraving for LM electronics. They studied about that using sophisticated material analyses. The reviewer thinks that the author needs to modify the manuscript in minor parts. Therefore, this article can be accepted in Bioengineering after minor revisions.

  1. In Figure 1b-d, the reviewer cannot recognized what the authors showed because of their low resolution without any caption in picture.
  2. In figure 5, the authors utilized the developed LM circuit for measuring the heart rate. Does LM circuit have biocompatibility? A simple cell toxicity test is needed to confirm this.
  3. The reviewer thinks that there are some biomedical applications which can use LM circuits. The reviewer recommends some state-of-arts and high-impact papers for adding in introduction.
  4. Advanced Materials35 (2020): 1907522. ( https://doi.org/10.1002/adma.201907522)
  5. Small Methods10 (2021): 2100762 ( DOI:10.1002/smtd.202100762 )
  6. Applied Surface Science520 (2020): 146304. ( https://doi.org/10.1016/j.apsusc.2020.146304 )

Author Response

Referee 2: In the article “Laser-engraved liquid metal circuit for wearable electronics”, the authors suggested novel patterning method using laser engraving for LM electronics. They studied about that using sophisticated material analyses. The reviewer thinks that the author needs to modify the manuscript in minor parts. Therefore, this article can be accepted in Bioengineering after minor revisions.

1. In Figure 1b-d, the reviewer cannot recognized what the authors showed because of their low resolution without any caption in picture. 

Reply: Many thanks to reviewer for their very professional advice. We are sorry that it is not clearly explained and hope that the corrections will meet with approval. We apologize for our poor data recording. We have increased the resolution of Figures 1, and added some caption in this picture.  

2. In figure 5, the authors utilized the developed LM circuit for measuring the heart rate. Does LM circuit have biocompatibility? A simple cell toxicity test is needed to confirm this. 

Reply: Thanks again to reviewer for their professional advice. Regarding your questions, we answer as follows: Gallium-based liquid metal has good biocompatibility and low cytotoxicity, and it could pass cytotoxicity tests.      

Skin cell of human were purchased and cultivated in glucose medium at 37 °C. Liquid metal nanodroplets (500 μg/mL) was used as culture medium for cultivation. Under a camera-equipped fluorescent microscope, the cytotoxicity could be evaluated by live or dead fluorescence observed. The experimental results show that liquid metal nanodroplet has good biocompatibility.

3. The reviewer thinks that there are some biomedical applications which can use LM circuits. The reviewer recommends some state-of-arts and high-impact papers for adding in introduction.  Advanced Materials35 (2020): 1907522. ( https://doi.org/10.1002/adma.201907522)Small Methods10 (2021): 2100762 ( DOI:10.1002/smtd.202100762 )Applied Surface Science520 (2020): 146304. ( https://doi.org/10.1016/j.apsusc.2020.146304 ) 

Reply: Thanks again to reviewer for their professional advice. Regarding your questions, we have revised these references, and added many high-impact and relevant references. And we have cited the relevant references in revised manuscript.     

Modified: Page 2, 12-13 in revised paper               

Laser technology has been widely used in various fields, for example: composite materials [39, 40], biomedical science [41-43], and so on. To the best of our knowledge, there have been few reports on the fabrication of ultra-micro LM electronic devices by laser engraving [44, 45].  

  1. Zhang, X.Y.; Stefan, P.; Pawel, R.; Malgorzata, M.; Dariusz, K.; Yang, J.L.; Thomas, G. Laser cladding of manganese oxide doped aluminum oxide granules on titanium alloy for biomedical applications, Applied Surface Science, 2020, 520, 146304-306.
  2. Jeong, Y.C.; Lee, H.E.; Shin, A.; Kim, D.G.; Lee, K.J.; Kim, D. Progress in Brain-Compatible Interfaces with Soft Nano-materials, Advanced Materials, 2020, 35, 1907522-38.
  3. Ma, W.; Li, J.; Li, X.J.; Bai, Y.; Liu, H.W.; Nanostructured Substrates as Matrices for Surface Assisted Laser Desorp-tion/Ionization Mass Spectrometry: A Progress Report from Material Research to Biomedical Applications, Small Methods, 2021, 10, 21007621-20.

Reviewer 3 Report

The research topic is modern and worth researching. The article requires a supplementation of the literature review in terms of the use of lasers in the textile industry, especially in the field of textronics applications.
e.g.
Korzeniewska, E .; Tomczyk, M .; Walczak, M. The Influence of Laser Modification on a Composite Substrate and the Resistance of Thin Layers Created Using the PVD Process. Sensors 2020, 20, 1920. https://doi.org/10.3390/s20071920
Buchman, A .; Rotel, M .; Dodiuk, H. Preadhesion surface laser treatment of composite, polymer and metal adherends. In Proceedings of the international conference on advanced composite materials (ICACM), Wollongong, Australia, 15-19 February 1993

Literature should be expanded to include items from the last 2 years. Some references are listed in a group like [12-14], [18-20]. Please provide more details of the works cited.

Fig. 5c-f should be enlarged because they are illegible in this study.

Part 2.2 and 2.3 have the same titles.

In my opinion, a list of the laser beam parameters should be included in the table.

The article lacks microscopic analysis of paths after mechanical bending. Does this layer not change its structure then?

Author Response

Referee 3: The research topic is modern and worth researching. The article requires a supplementation of the literature review in terms of the use of lasers in the textile industry, especially in the field of textronics applications.    

1. e.g. Korzeniewska, E .; Tomczyk, M .; Walczak, M. The Influence of Laser Modification on a Composite Substrate and the Resistance of Thin Layers Created Using the PVD Process. Sensors 2020, 20, 1920. https://doi.org/10.3390/s20071920Buchman, A .; Rotel, M .; Dodiuk, H. Preadhesion surface laser treatment of composite, polymer and metal adherends. In Proceedings of the international conference on advanced composite materials (ICACM), Wollongong, Australia, 15-19 February 1993Literature should be expanded to include items from the last 2 years. Some references are listed in a group like [12-14], [18-20]. Please provide more details of the works cited. 

Reply:  Thanks for your kind advice. We have added more recent references in terms of the use of lasers in the textile industry and textronics applications. And we have cited the relevant references from the last 2 years in revised manuscript. And we have revised the order of the references, and provide more details of the works cited.       

Modified: Page 2, 12-13 in revised paper               

       Laser technology has been widely used in various fields, for example: textronics [39], composite materials [40], biomedical science [41-43], and so on. To the best of our knowledge, there have been few reports on the fabrication of ultra-micro LM electronic devices by laser engraving [44, 45].  

39. Ewa, K.; Mariusz, T.; Maria, W. The Influence of Laser Modification on a Composite Substrate and the Resistance of Thin Layers Created Using the PVD Process, Sensors 2020, 20, 1920.

40. Alisa, B.; Rotel, M.; Zahavi, J.; Dodiuk, H. Preadhesion surface laser treatment of composite, polymer and metal adherends IMEC VI, In Proceedings of the international conference on advanced composite materials (ICACM), Wollongong, Australia, 1993.  

2. Fig. 5c-f should be enlarged because they are illegible in this study. 

Reply: Thanks for your kind advice. We have enlarged c-f in Figure 5 in the manuscript.           

The revised version is as follows:      

3. Part 2.2 and 2.3 have the same titles.  

Reply: Many thanks to reviewer for their professional advice. We have revised the title accordingly. The revised version is as follows:    

Original: Page 3 in manuscript   

2.2. Laser engraving  

2.3. Laser engraving    

Modified: Page 3 in rvised paper         

2.2. Laser engraving process      

2.3. Electrical/Morphological Characterization      

4. In my opinion, a list of the laser beam parameters should be included in the table.  

Reply: Thanks for your comments. We have rewritten and added a list of the laser beam parameters in a table, and appended it to the Supporting Information.   

The revised version is as follows:    

Modified: Page 1 in Supporting Information          

Table R1 The relationship between carving speed, travel speed, carving time and engraving width in the sample

Sample

Carving speed

(mm/s)

travel speed

(mm/s)

Carving time

(s)

Engraving width

(μm)

Sample 1

1.667

5

18

650

Sample 2

8.333

5

3

253

Sample 3

8.333

5

6

300

Sample 4

8.333

11.667

6

200

5. The article lacks microscopic analysis of paths after mechanical bending. Does this layer not change its structure then? 

Reply: Thanks for your kind advice. That is a very good question, we used liquid metal as the bending material in our experiment.

As could be seen from figure 3a, the liquid metal circuit has a depth of 200 μm, which means that there is a lot of liquid metal still left inside the PDMS groove when mechanical bending occurs.

After mechanical bending, because liquid metal is liquid, its morphology and structure would not change greatly in the process of mechanical bending. 

Round 2

Reviewer 1 Report

There is an error with Ref : 39 --- Ewa, Mariusz and Maria are first names not surnames

Ewa, K.; Mariusz, T.; Maria, W. The Influence of Laser Modification on a Composite Substrate and the Resistance of Thin Layers 513
Created Using the PVD Process, Sensors 2020, 20, 1920.

Author Response

Referee 1:  There is an error with Ref : 39 --- Ewa, Mariusz and Maria are first names not surnames.  Ewa, K.; Mariusz, T.; Maria, W. The Influence of Laser Modification on a Composite Substrate and the Resistance of Thin Layers 513Created Using the PVD Process, Sensors 2020, 20, 1920. 

Reply: Thank you very much for your suggestions. We have revised the reference [39].

Original: Page 13 in manuscript   

39. Ewa, K.; Mariusz, T.; Maria, W. The Influence of Laser Modification on a Composite Substrate and the Resistance of Thin Layers Created Using the PVD Process, Sensors 2020, 20, 1920.

Modified: Page 13 in revised paper               

39. Korzeniewska, E.; Tomczyk, M.; Walczak, M. The Influence of Laser Modification on a Composite Substrate and the Resistance of Thin Layers Created Using the PVD Process, Sensors 2020, 20, 1920.  

Reviewer 3 Report

In added references, the surnames are changed with the names of the authors. please check it. Dariusz,  Mariusz, Maria, Ewa, MaÅ‚gorzata are the Polish names. So in my opinion, the rest names are not correctly cited.

Author Response

     In added references, the surnames are changed with the names of the authors. please check it. Dariusz, Mariusz, Maria, Ewa, MaÅ‚gorzata are the Polish names. So in my opinion, the rest names are not correctly cited.

Reply: Many thanks to reviewer for their very professional advice. We have revised these references. 

Original: Page 13 in manuscript    

40. Alisa, B.; Rotel, M.; Zahavi, J.; Dodiuk, H. Preadhesion surface laser treatment of composite, polymer and metal adherends IMEC VI, In Proceedings of the international conference on advanced composite materials (ICACM), Wollongong, Australia, 1993.

41. Zhang, X.Y.; Stefan, P.; Pawel, R.; Malgorzata, M.; Dariusz, K.; Yang, J.L.; Thomas, G. Laser cladding of manganese oxide doped aluminum oxide granules on titanium alloy for biomedical applications, Applied Surface Science, 2020, 520, 146304-306.   

Modified: Page 13 in revised paper     

40. Dodiuk, H.; Buchman, A.; Rotel, M.; Zahavi, J. Preadhesion surface laser treatment of composite, polymer and metal adherends IMEC VI, In Proceedings of the international conference on advanced composite materials (ICACM), Wollongong, Australia, 1993.

41. Zhang, X.Y.; Pfeiffer, S.; Rutkowski, P.; Makowska, M.; Kata, D.; Yang, J.L.; Graule, T. Laser cladding of manganese oxide doped aluminum oxide granules on titanium alloy for biomedical applications, Applied Surface Science, 2020, 520, 146304-306.